# Cloning and Characterization of Three Sugar Metabolism Genes (*LBGAE*, *LBGALA*, and *LBMS*) Regulated in Response to Elevated CO_2_ in Goji Berry (*Lycium barbarum* L.)

**DOI:** 10.3390/plants10020321

**Published:** 2021-02-07

**Authors:** Yaping Ma, Mura Jyostna Devi, Vangimalla R. Reddy, Lihua Song, Handong Gao, Bing Cao

**Affiliations:** 1College of Forestry, Nanjing Forestry University, Nanjing 210037, China; YapingMa@njfu.edu.cn (Y.M.); gaohd@njfu.edu.cn (H.G.); 2School of Agriculture, Ningxia University, Yinchuan 750021, China; slh382@126.com; 3USDA-ARS, Adaptive Cropping Systems Laboratory, 10300 Baltimore Ave, Beltsville, MD 20705, USA; Vangimalla.Reddy@ars.usda.gov; 4USDA-ARS, Vegetable Crops Research Unit, Madison, WI 53706, USA; 5Department of Horticulture, University of Wisconsin-Madison, Madison, WI 53705, USA

**Keywords:** goji berry, sugar metabolism, elevated CO_2_, functional domain, gene cloning and expression

## Abstract

The composition and content of sugar play a pivotal role in goji berry (*Lycium barbarum* L.) fruits, determining fruit quality. Long-term exposure of goji berry to elevated CO_2_ (eCO_2_) was frequently demonstrated to reduce sugar content and secondary metabolites. In order to understand the regulatory mechanisms and improve the quality of fruit in the changing climate, it is essential to characterize sugar metabolism genes that respond to eCO_2_. The objectives of this study were to clone full-length cDNA of three sugar metabolism genes—*LBGAE* (*Lycium barbarum* UDP-glucuronate 4-epimerase), *LBGALA* (*Lycium barbarum* alpha-galactosidase), and *LBMS* (*Lycium barbarum* malate synthase)—that were previously identified responding to eCO_2_, and to analyze sequence characteristics and expression regulation patterns. Sugar metabolism enzymes regulated by these genes were also estimated along with various carbohydrates from goji berry fruits grown under ambient (400 μmol mol^−1^) and elevated (700 μmol mol^−1^) CO_2_ for 90 and 120 days. Homology-based sequence analysis revealed that the protein-contained functional domains are similar to sugar transport regulation and had a high sequence homology with other Solanaceae species. The sucrose metabolism-related enzyme’s activity varied significantly from ambient to eCO_2_ in 90-day and 120-day samples along with sugars. This study provides fundamental information on sugar metabolism genes to eCO_2_ in goji berry to enhance fruit quality to climate change.

## 1. Introduction

Goji berry (*Lycium barbarum*) is a perennial shrub and belongs to the genus *Lycium* of the family Solanaceae, also called wolfberry. China is the leading supplier of goji, with a yield of 95,000 tons and cultivating area of nearly 90,000 hectares. Currently, the goji berry is being used worldwide as popular nutraceutical food or dietary health supplement in various forms due to its rich chemical composition and healing properties [1]. With the increasing demand due to its medicinal properties, more attention has been paid to improve the fruit quality of the goji berry. The LBP (*Lycium barbarum* polysaccharide) is the most abundant group of active ingredients in goji berry fruit, containing several monosaccharides that are influenced by several environmental factors [2,3].

Climate change has profound global consequences with direct effects on agricultural production, livestock and environment quality, and indirect influence on food security and human survival [4]. Elevated atmospheric CO_2_ concentration is beneficial for photosynthesis and can directly affect the growth and development of plants [5,6,7]. While plants exposed to increased levels of CO_2_ are expected to benefit from carbon fertilization, the extent of advantage depends on the plant species. For example, in tomato, carbon exchange rates were significantly higher in CO_2_-enriched plants for the first few weeks of treatment after that decreased due to acclimation to elevated CO_2_, which attributed to an accumulation of sugar and/or starch [8]. Similarly, prolonged elevated atmospheric CO_2_ levels were shown to negatively affect goji berry fruit quality with reduced flavonoid, carotenoid, total sugars and polysaccharides, which might be related to photosynthesis acclimation [9].

A long-term elevated CO_2_ condition accelerates the vegetative growth and improves the morphology of the fruit, but it was also found that the contents of fructose, glucose, sucrose, polysaccharide, and total sugar were reduced. The sucrose-metabolizing enzymes changed significantly over the period across treatments [10]. Similar results were also observed in goji berry fruits grown under eCO_2_, where the reduction in sugar and secondary metabolite levels was noticed, affecting fruit quality [9]. The sugar content in the fruit is determined by the sugar biosynthesis and accumulation, which is catalyzed by relating enzymes controlled by the expression of genes. The reduction in the sink and an increase in non-structural carbohydrate levels due to photosynthesis acclimation were related to a single gene mutation in soybean [11]. Hence, it is important to identify and characterize the genes involved in sugar metabolism to improve the quality of the goji berry fruit for future climatic conditions, especially with the projected increase in the atmospheric CO_2_ concentration.

Sucrose, glucose, and fructose are the main components of plant fruits. Sucrose is a significant form of photosynthate component transportation from the source (leaves) and is distributed to sink organs (fruit). Sucrose supplies carbon and energy to plants by first relying on their cleavage to form hexoses. This process is catalyzed by at least two different classes of enzymes: one is sucrose synthase, which catalyzes the cleavage of sucrose to uridine diphosphate (UDP) and fructose; the other is invertase, catalyzing the cleavage of sucrose to form glucose and fructose [12,13,14,15]. The molecular regulation of sugar metabolism and genes involved in such regulation in goji berry fruit to different environmental conditions is still unknown, especially under elevated CO_2_. Hence, the objective of the current study was to clone and analyze the sequence of three goji berry genes—*LBGAE* (*Lycium barbarum* UDP-glucuronate 4-epimerase), *LBGALA* (*Lycium barbarum* alpha-galactosidase), and *LBMS* (*Lycium barbarum* malate synthase)—that were identified in our previous study through transcriptome profiling [9]. The reason for selecting these three genes was that they were upregulated under eCO_2_ and are involved in four primary sugar metabolic pathways. The three genes involved in sugar metabolism pathways and regulation, especially in sucrose metabolism, did not express under ambient conditions but increased their expression under elevated CO_2_ treatment. The other objective was to study the activity of four sucrose metabolism-related enzymes (sucrose phosphate synthase, neutral invertase, acid invertase, sucrose synthase) regulated by the three genes. The end components of these three genes reducing sugars, starch, polysaccharide, and sucrose, were also estimated to observe the effects of elevated CO_2_ on substance accumulation and metabolism in goji berry to improve the fruit quality.

## 2. Results

### 2.1. Identification and the Complete Sequence Cloning of Sugar Metabolism Genes

Three sugar metabolism-related genes *LBGAE* (*Lycium barbarum* UDP glucuronate 4-epimerase), *LBGALA* (*Lycium barbarum* alpha-galactosidase) and *LBMS* (*Lycium barbarum* malate synthase) upregulated to elevated CO_2_ were selected from a previous transcriptome profiling study of goji berry [9]. These three genes were upregulated under eCO_2_ and are involved in four primary sugar metabolic pathways. Subsequently, this study characterized the three genes since these genes are involved in four sugar pathways. The sequences obtained by the transcriptome profiling are usually incomplete; hence, cloning of cDNA of *LBGAE*, *LBGALA,* and *LBMS* was performed to analyze the complete sequence.

The full-length cDNA fragments of *LBGAE* (GenBank accession No. MH025911), *LBGALA* (GenBank accession No. MH025913), and *LBMS* (GenBank accession No. MH025912) obtained by RACE-PCR (rapid amplification of cDNA) were 1837 bp, 1233 bp, and 2449 bp, respectively. The *LBGAE* contains a 5′-untranslated region (UTR) of 466 bp and an open reading frame (ORF) of 1236 bp encoding 411 apiece, and a 3′-untranslated region (UTR) of 135 bp. Similarly, the *LBMS* contains a 5′- UTR of 400 bp and an ORF of 1716 bp encoding 571 amino acids apiece, and a 3′- UTR of 333 bp. The *LBGALA* only contained an ORF of 1233 bp encoding 410 amino acids. The nucleotide and deduced amino acid sequences of *LBGAE*, *LBGALA,* and *LBMS* are shown in Appendix A.

### 2.2. Characterization and Sequences Analysis

Protein sequence analysis demonstrated that *LBGAE*, *LBGALA*, and *LBMS* contain a conservative structure domain of SDR (short-chain dehydrogenases/reductases; GenBank: CL25409, position: 73–401), AmyAc (Alpha-amylase catalytic; GenBank: cl07893, position: 38–409), and malate synthase (GenBank: PLN02626, position: 19–571), respectively (Table 1). The physicochemical properties of proteins, such as molecular weight, theoretical pI, instability index, aliphatic index, and grand average hydropathicity, were analyzed and listed in Table 1. In addition, the analysis indicated that the *LBGAE* protein sequence contains a transmembrane region (position: 7–29) and six protein functional domains (Figure 1a). The *LBGALA* protein sequence includes signal peptide cleavage sites in Ala^23^ and two protein functional domains of glycoside hydrolase family (Figure 1b). The analysis also indicated that *LBGALA* is a secretory protein, while *LBGAE* and *LBMS* are soluble and non-secretory proteins synthesized in the cytoplasm without protein translocation ability.

Based on protein solvent accessibility composition (core and surface ration) analysis, the ration of residues exposed on the protein surface of *LBGAE*, *LBGALA*, and *LBMS* was 45.74%, 43.41%, and 44.13%, respectively. The residue ration in the protein core of *LBGAE*, *LBGALA,* and *LBMS* was 54.26%, 56.59% and 55.87%. This confirms that these three genes’ hydrophilicity is weaker than that of hydrophobicity. The analysis showed different protein sites in the protein sequence (details are shown in Appendix A). Similarly, the phosphorylation site prediction of the *LBGAE*, *LBGALA*, and *LBMS* protein sequences reveals phosphorylation sites of 44 (24 serine, 12 threonines, 8 tyrosine), 45 (25 serine, 14 threonines, 6 tyrosine), and 26 (12 serine, 7 threonines, 7 tyrosine), respectively, for these genes.

The subcellular localization prediction discloses that *LBGAE*, *LBGALA*, and *LBMS* have the highest possibility of located in mitochondria, chloroplast, and peroxisomes (Table 1). The prediction of protein secondary (Appendix A) and tertiary structures (Figure 2a–c) indicated that *LBGAE*, *LBGALA*, and *LBMS* are mainly composed of alpha-helix, random coil and extended strand (Table 1). Additionally, no disulfide bond was formed among them.

### 2.3. Multiple Sequence Alignment and Phylogenetic Analysis

The multiple sequence alignment of *LBGAE*, *LBGALA,* and *LBMS* with amino acid sequences of several other species from NCBI was performed using ClustalX and MEGA 7.0 program. Results indicated that three genes are more than 80% identical to the *Capsicum annuum*, *Solanum tuberosum*, *Solanum lycopersicum* and *Nicotiana tabacum* (Appendix A).

The phylogenetic tree was constructed with full-length proteins of the three genes showed that these three genes were clustered with other homologous genes of 20 species into three groups. The three genes were clustered in the same branch with members of *Nicotiana*, *Capsicum* and *Solanum*, but distantly related with members of *Momordica charantia*, *Erythranthe guttata*, and *Arabidopsis thaliana* (Figure 3).

### 2.4. Regulation Pathways of the Genes LBGAE, LBGALA and LBMS

*LBGAE*, *LBGALA* and *LBMS* genes in different pathways and different stages of sugar conversion are detailed in Figure 4. These three genes are involved in the regulation of four sugar metabolism pathways: amino sugar and nucleotide sugar metabolism (ko00520), starch and sucrose metabolism (ko00500), pyruvate metabolism (ko00620), and galactose metabolism (ko00052) (Figure 4). The gene *LBGAE* (EC: 5.1.3.6, UDP-glucuronate 4-epimerase) is mainly responsible for the interconversion of UDP-D-Galacturonate and UDP-D-Glucuronate in starch and sucrose metabolism (ko00500), and UDP-GlcA and UDP-Gala in amino sugar and nucleotide sugar metabolism (ko00520) as represented in Figure 4.

In galactose metabolism (ko00052), *LBGALA* (EC: 3.2.1.22, alpha-galactosidase) acts at several places, including the conversion of stachyose, mannotriose, melibiose, epimelibiose and D-mannose, as well as glycerol and galactosyl-glycerol; it also converts raffinose into sucrose and stachyose. *LBGALA* is also involved in the regulation of lipid metabolism (K00561). Similarly, *LBMS* is involved in glyoxylate and dicarboxylate metabolism (K00630) and pyruvate metabolism (ko00620). The MS (EC: 2.3.3.9, malate synthase) is primarily responsible for the conversion of Acetyl-CoA to L(S)-Malate (Figure 4).

### 2.5. Expression Analysis of Genes LBGAE, LBGALA, and LBMS

Further evaluation of *LBGAE*, *LBGALA*, and *LBMS* to eCO_2_ by qRT-PCR was not consistent among leaves, stems, and fruits. The expression levels of the three genes in stems and leaves measured at 90 and 120 days indicated that the expression of *LBGAE* and *LBGALA* was significantly higher under ambient conditions at 120 days in stems (Figure 5e). In fruits, three genes’ expression levels were significantly higher under ambient conditions at 90 days (Figure 5c). However, the same genes showed higher expression under elevated CO_2_ than ambient conditions at 120 days. Among them, the expression of *LBGAE* was significantly higher than *LBGALA* and *LBMS* (*p* < 0.05, *p* < 0.01) (Figure 5).

### 2.6. Sugar Content and Sucrose Metabolism-Related Enzymes Activities

Various sugar components and sucrose metabolism-related enzymes in goji berry fruits grown under ambient and elevated CO_2_ for 90 and 120 days were determined to understand three genes’ role in sugar metabolism (Figure 6). All four estimated sugars, including reducing sugar, starch, polysaccharide, and sucrose, followed a downward trend under elevated CO_2_. The levels of reducing sugar and starch decreased significantly in fruit samples grown under eCO_2_ at both 90 and 120 days (Figure 6a,b), while polysaccharide and sucrose declined only in 120-day samples (Figure 6c,d) (*p* < 0.05), compared to ambient CO_2_ conditions.

The activity of four sucrose metabolism-related enzymes, sucrose phosphate synthase, neutral invertase, acid invertase, and sucrose synthase were also determined in the samples collected at 90 and 120 days. Fruits grown under elevated CO_2_ for 90 days showed a significant increase in the activity of four enzymes (Figure 6e–h) (*p* < 0.01). However, the activity of sucrose phosphate synthase and sucrose synthase decreased significantly in the samples collected at 120 days (Figure 6e,h) (*p* < 0.01) compared to ambient CO_2_ levels.

## 3. Discussion

### 3.1. Cloning and Sequence Analysis of Sugar Metabolism Genes

The current study isolated three sugar metabolism genes *LBGAE*, *LBGALA,* and *LBMS* from goji berry fruit under elevated CO_2_ and analyzed their sequence characteristics and potential functions. The gene *LBGAE* has 1236 bp ORF that encodes a 411-amino acid polypeptide, a putative protein containing SDR (short-chain dehydrogenases/reductases) superfamily conserved domain. It constitutes a large family of NAD(P)(H)-dependent oxidoreductases and a structurally diverse C-terminal region, with enzymes having critical roles in carbohydrate, lipid, and amino acid metabolism [16]. The gene *LBGALA* has 1233 bp ORF that encodes a 410 amino acid polypeptide. The *LBGALA* putative protein includes an Amy-Ac (Alpha-amylase catalytic) domain family and is the largest family of glycoside hydrolases (GH), usually acting on starch, glycogen, related oligosaccharides, and polysaccharides. The enzyme Amy-Ac can catalyze the conversion of alpha-1,4 and alpha-1,6-glucose chains [17]. Similarly, *LBMS* has 1716 bp ORF that encodes a 571 amino acid polypeptide, the putative protein with a malate synthase conserved domain. The malate synthase catalyzes the Claisen condensation of glyoxylate and acetyl-CoA to malyl-CoA, which hydrolyzes to malate and CoA. In addition, malate is also involved in the glyoxylate cycle that allows certain organisms, such as plants and fungi, to derive their carbon requirements from two carbon compounds by bypassing the two carboxylation steps of the citric acid cycle [18]. These three genes influence the sucrose metabolism-related enzymes sucrose phosphate synthase, neutral invertase, acid invertase, and sucrose synthase estimated in response to eCO_2_ in this study (see the schematic representation of sugar metabolism in Figure 4) and strongly suggest the involvement of *LBGAE*, *LBGALA*, and *LBMS* in the regulation of sugar metabolism under elevated CO_2_. Phylogenetic alignment of amino acid sequences revealed a very high sequence homology of putative *LBGAE*, *LBGALA*, and *LBMS* sequences with *Capsicum annuum*, *Solanum tuberosum*, *Solanum lycopersicum,* and *Nicotiana tabacum* of Solanaceae family proteins.

Overall, the sequence analysis of these three genes revealed that the genes’ protein structures contain some functional domains, which provide initial evidence of their involvement in sugar metabolism, and this information can be used to explore sugar metabolism to eCO_2_.

### 3.2. Regulation and Expression

The schematic representation of the transcriptome data of the three genes [9] in sugar metabolism pathways shows genes involved in four sugar metabolism pathways and their overexpression under elevated CO_2_ in fruits (Figure 4 and Figure 5). Along with sugar metabolism, *LBGALA* displayed an up-regulation in the other two lipid metabolism pathways (Figure 4). These genes amplified their expression levels to eCO_2_ treatment compared to ambient conditions in fruits at 120 days (Figure 5). However, gene transcripts measured in leaf and stem were different from the results of the fruits. Mostly, the expression of these genes in eCO_2_ treatment was significantly lower or not different than ambient CO_2_ conditions both at 90 and 120 days in leaf and stem samples. Elevated atmospheric CO_2_ concentration in the long-term is known to affect the photosynthesis and its components as a response to increased carbohydrate production and metabolism. Long-term exposure to elevated CO_2_ can result in reduced levels of mRNAs encoding specific photosynthesis genes due to increased glucose or sucrose levels in the leaves [5,19,20]. The differential expression of the sugar metabolism genes, hexokinase and sucrose phosphate synthase, was found to play a role in exerting regulatory influence on sucrose biosynthesis to accommodate enhanced photosynthesis in *Jatropha* subjected to a CO_2_-enriched environment [5]. In contrast, the variation in leaf soluble carbohydrate amount at elevated and ambient CO_2_ concentrations reduced with crop development, whereas the difference in transcript levels increased [21]. In this study, the reduction in the levels of the three genes under eCO_2_ over the ambient CO_2_ conditions in leaves and stems might be related to increased sucrose and glucose levels in the leaves [19]. The increase in the levels of the genes expressed in the fruit to elevated CO_2_ could be a feedback regulation to compensate for reduced sugar components. These results demonstrate that three genes’ potential regulatory function in the sugar metabolism of goji berry influences enzyme activity and metabolism.

### 3.3. Response of Sugar and Sucrose Metabolism-Related Enzymes to Elevated CO_2_

In previous studies with goji berry, CO_2_ enrichment showed an influence on growth, photosynthesis, flavonoids, carotenoids, and sugars [9,22]. Enhanced CO_2_, especially in C3 plants, causes increased photosynthesis, which generally leads to increased production of carbohydrates [23]. However, in the present study, the estimated four different kinds of sugars showed a reduction under eCO_2_ in samples treated for 90 and 120 days compared to ambient conditions. The results agree with other studies that the elevated CO_2_ reduces the sugar content in goji berry fruits [9,22,24]. Some studies demonstrated an increase in reducing sugars, total sugar, and starch content under short-term eCO_2_ (850 ppm) compared to ambient conditions. However, long-term exposure reduces their levels, possibly due to photosynthesis acclimation [25]. Other studies observed higher carbon exchange rates to eCO_2_ in the first few weeks of treatment but later decreased due to photosynthesis acclimation and low sink source or high sugar accumulation [8,11]. The possible explanation for the reduction in the four sugar components estimated is photosynthesis acclimation. However, no significant differences in the sugars were observed between 90 and 120 days, which suggests the occurrence of photosynthesis acclimation before 90 days. Further studies measuring photosynthesis along with sugar levels at different developmental stages are required to understand the process of photosynthesis acclimation in goji berry.

Sucrose, glucose, and fructose are the main soluble sugars in the fruit of the plant, and the contents of these sugars play a key role in the quality of fruit. Research in goji berry and other fruit crops indicated a close relationship between fruit sugar accumulation and sucrose-metabolizing enzymes, including sucrose synthase, sucrose phosphate synthase, acid invertase, and neutral invertase [12,26,27]. In contrast to the sugar components response, a significant variation from 90 days to 120 days in the activity levels of sucrose metabolism-related enzymes was observed (Figure 6e–h). The activity levels were higher in samples of eCO_2_ than ambient CO_2_ at 90 days. However, a significant drop at 120 days, which might be related to a reduction in the sugar levels was observed. These results agree with previous studies where sucrose metabolism-related enzymes were affected; sugar and secondary metabolite levels were reduced [10,28]. In a study with *Arabidopsis*, overexpression of sucrose phosphate synthase increased foliar sucrose/starch ratios and decreased foliar carbohydrate contents when plants were grown for long periods under enriched CO_2_ conditions [29]. In rice, a high leaf CO_2_ exchange rate was observed to enhanced CO_2_ resulting in increased peduncle exudate sucrose levels but decreased levels in the developing grains. The whole process was linked to the poor responses to enhanced CO_2_ in the activity of sucrose synthase, UDP-glucose pyrophosphorylase, ADP-glucose pyrophosphorylase, and starch synthases enzymes that are involved in the conversion of sucrose to starch [30]. Overall, this study demonstrates that the elevated CO_2_ levels affected the enzyme involved in the sucrose metabolism and quality of goji berry fruit by reducing its sugar components.

## 4. Materials and Methods

### 4.1. Plant Material and Elevated CO_2_ Treatment

The cultivar used in the present study was “Ningqi NO.1”, and experiments were conducted with a one-year-old goji berry seedling obtained from the Ningxia Academy of Agriculture and Forestry Science, China. Experiments were conducted in open-top chambers (OTCs) [31,32] with two treatments ambient (400 μmol·mol^−1^ CO_2_) and elevated CO_2_ (700 μmol·mol^−1^ CO_2_) for 120 days. The experimental site is located in Yongning county of yellow river alluvial plain in Central Ningxia of China (38°13′50.34″ N; 106°14′22.19″ E, 1116.86 m above sea level, located in the northwest inland). The experimental farm’s climatic conditions are arid and middle temperate with a frost-free period of 140–160 days. The annual sunshine duration in the location was 3000 h, with an average annual temperature of 8.5 °C and precipitation of 180~300 mm. Matured fruit samples (red fruit after the color change period) were selected from the well-grown (treatment 120 days) and similar flowering period plants for the analysis. The collected samples were frozen immediately in liquid nitrogen and stored at −80 °C for further analysis.

### 4.2. Extraction of Total RNA and cDNA Synthesis

Total RNA was extracted from the fruits of goji berry by using RNAprep Pure Plant Kit (Tiangen, Beijing, China), according to the manufacturer’s protocol. RNA concentration was measured using a NanoDrop 2000 spectrophotometer (Thermo Scientific, Waltham, MA, USA), and the quality was assessed by performing electrophoresis on 0.8% agarose gel. First-strand cDNA synthesis was synthesized using the PrimeScript™RT Reagent Kit with gDNA Eraser (Takara, Dalian, China), and the samples were stored at −20 °C until sequenced.

### 4.3. Cloning Analyses of the LBGAE, LBGALA and LBMS

The sequences of *LBGAE*, *LBGALA*, and *LBMS* obtained from transcriptome sequencing were searched in the NCBI database. Based on the search, the sequences of *LBGAE* and *LBGALA* were incomplete, and the *LBMS* gene has a complete sequence. Rapid amplification of cDNA ends (RACE) was used to obtain the full-length cDNA sequence of *LBGAE* and *LBGALA* genes. The 3′ and 5′ RACE reactions were carried out using a SMARTer^®^ RACE 5′/3′ Kit (Clontech, Beijing, China) following the manufacturer’s recommendations. The RACE primers were synthesized using SNAPGENE and primer 5.0. The specific primers 3′-RACE (GSP1) and 5′-RACE (GSP2) were designed to be specific to the target gene (Appendix A).

A primer pair, F2 and R2 (Appendix A), was designed by employing PrimeSTAR Max DNA (Takara, Dalian, China) to amplify the full length of *LBGAE* and *LBGALA*. The steps of PCR amplification include a pre-denaturation at 94 °C for 5 min, followed by 33 cycles of 94 °C for 40 s, 58 °C for 30 s, 72 °C for 1 min 20 s, and then a final extension at 72 °C for 10 min and stored at 16 °C. Products of PCR were visualized and verified using 0.8% agarose gel. The product was then extracted and purified utilizing a PCR Purification Kit (Tiangen, Beijing, China), and then ligated into the pMD18-T vector (Takara, Dalian, China) and sequenced.

### 4.4. Bioinformatics Analysis of Sequences

DNAMAN (V8.0) software was used to evaluate the molecular mass, base composition, and base distribution of the nucleic acid sequences. In addition, BioEdit (V7.0.5.3) software was used to perform sequence analysis of the enzyme digestion. Base-homology analysis was conducted utilizing BLAST of NCBI (http://www.ncbi.nlm.nih.gov/BLAST/) (accessed on 5 January 2021), and protein structural domain analysis of the protein sequence was studied employing SMART (Simple Modular Architecture Research Tool) (http://smart.embl-heidelberg.de/) (accessed on 5 January 2021) tool. The open reading frame was identified by ORF finder (https://www.ncbi.nlm.nih.gov/orffinder/) (accessed on 5 January 2021). Protein tertiary structure was modeled by the homology-modeling server SWISS-MODEL (http://swissmodel.expasy.org/) (accessed on 5 January 2021), and the models were further analyzed and refined using Swiss-Pdb Viewer v4.1.0.

The molecular weight and isoelectric point (pI) of protein were estimated by ProtParam (http://web.expasy.org/protparam/) (accessed on 5 January 2021). Multiple sequence alignment and phylogenetic analysis were performed using ClustalX 2.0, MEGA 7.0. Software tools PSort II (http://psort.hgc.jp/) (accessed on 5 January 2021), TMHMM (http://www.cbs.dtu.dk/services/TMHMM/) (accessed on 5 January 2021), and SignalP4.1 (http://www.cbs.dtu.dk/services/SignalP/) (accessed on 5 January 2021) were employed to predict the protein subcellular location, the single transmembrane region, and the signal peptide, respectively. Protein hydrophobicity and hydrophilicity were assessed through Protscale (http://web.expasy.org/protscale/) (accessed on 5 January 2021). A phylogenetic tree of proteins from different species was constructed using the neighbor-joining (NJ) method in the MEGA7.0 software (https://www.megasoftware.net) (accessed on 5 January 2021). The reliability of the tree was measured by bootstrap analysis with 1000 replications.

### 4.5. Gene Expression Analysis by qRT-PCR

The specific primers for qPCR of *LBGAE*, *LBGALA*, and *LBMS* and *β*-actin (internal reference) were designed based on the full cDNA sequence (Appendix A), and the primers were designed using PrimerQuest Tool (http://sg.idtdna.com/Primerquest/Home/Index) (accessed on 5 January 2021). The qRT-PCR assay was conducted using a Quantitative Fluorescent assay that used the abm^®^EvaGreen qPCR MasterMix-ROX kit (ABM, Vancouver, BC, Canada).

The qRT-PCR was performed with a 20 µL of reaction mix containing 10 µL abm^®^EvaGreen qPCR MasterMix-No Dye (2×), 0.4 µL of each Primer (10 μM, 0.8 µL of PCR Reverse Primer), 10 μM, 1.0 µL of cDNA template, and 7.4 µL of sterilized ddH_2_O. The cycle of qRT-PCR reaction program was denaturation at 95 °C for 10 min, followed by 45 cycles of 94 °C for 15 s, 60 °C for 1 min, and stored at 16 °C. The qPCR assays were performed by using ABI StepOne Plus (Applied Biosystems, Foster City, USA) with nine biological and three technical replicates. The relative expression levels of *LBGAE*, *LBGALA*, and *LBMS* were calculated using the 2^−ΔΔct^ method [33].

### 4.6. Polysaccharide, Sucrose, Reducing Sugar, Starch, and Sucrose Metabolism-Related Enzymes Estimation

Fruit samples were collected in nine replicates from ambient and elevated CO_2_ OTC chambers on 90 and 120 days. The samples were stored at −80 °C until further the next stage. The content of polysaccharide, sucrose, reducing sugars, and starch was measured by following the method described by Ha et al. [10] using approximately 5.0 g sample. The enzyme activity of sucrose phosphate synthase, neutral invertase, acid invertase, and sucrose synthase was estimated as described by Liu et al. [28].

## 5. Conclusions

This study confirms the effect of eCO_2_ on various sugar components and enzyme activities of goji berry fruit estimated at both 90 and 120 days. Furthermore, the three sugar metabolism genes that upregulated to eCO_2_ were analyzed in detail to understand the molecular mechanism of the goji berry response to eCO_2_. The full-length cDNAs of *LBGAE*, *LBGALA*, and *LBMS* were cloned and analyzed for their protein characteristics. The potential regulatory functions of the proteins were identified; in addition, significant upregulation transcriptome of these three genes in fruits to CO_2_ was confirmed. The expression of these genes might have played an important role in the sucrose-metabolizing enzyme activity and reduced the content of four sugars estimated. However, further studies are required to understand the underlying regulatory mechanism of these genes to eCO_2_. The results of this study provide theoretical references and lay the molecular basis to study metabolic regulation of sugar accumulation and the subsequent understanding of gene functions in goji berry to improve fruit quality in changing climate conditions. However, a detailed study on the role of three genes and their mechanism in regulating sugar metabolism by knock-out or knock-down mutants is needed.

## Figures and Tables

**Figure 1 plants-10-00321-f001:**
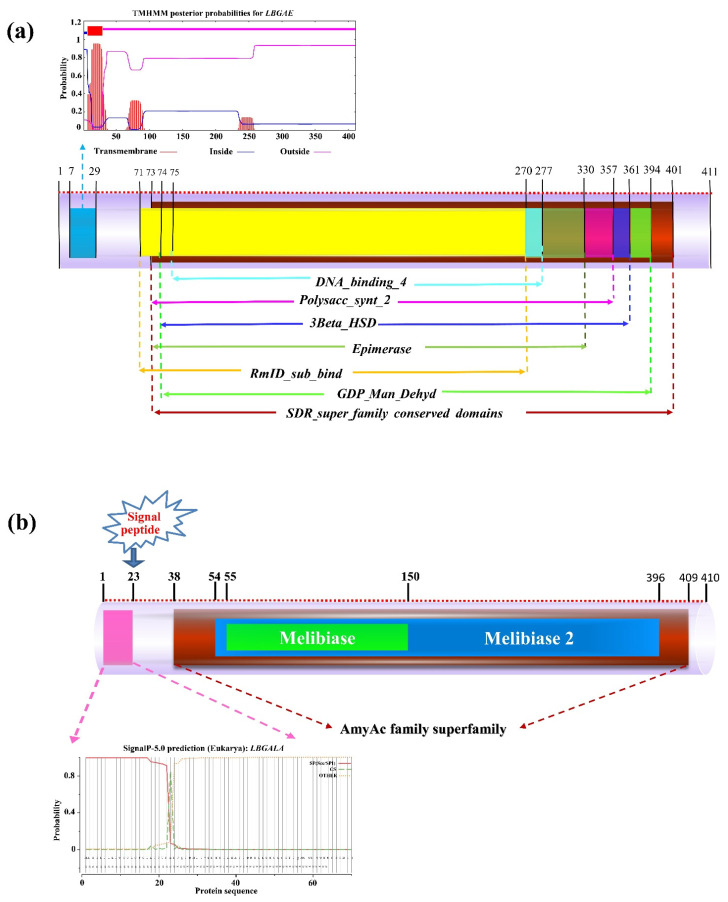
Protein functional domain analysis of *LBGAE* (**a**) and *LBGALA* (**b**). The *LBGAE*, including a *RmID_sub_bind* (RmID substrate binding domain, at position 71–270), an *Epimerase* (NAD-dependent epimerase/dehydratase family, 73–330), a *Polysacc_synt_2* (polysaccharide biosynthesis protein, 73–357), a *GDP_Man_Dehyd* (GDP-mannose4,6 dehydratase, 74–394), a *3Beta_HSD* (3-beta-hydroxysteroid dehydrogenase/isomerase family, 74–361), and a DNA*_binding_4* (male sterility protein, 75–277). The transmembrane region (position: 7–29) and the *LBGALA* including a *Melibiase_2* (54–396) and *Melibiase* (55–150). A signal peptide cleavage site in Ala^23^.

**Figure 2 plants-10-00321-f002:**
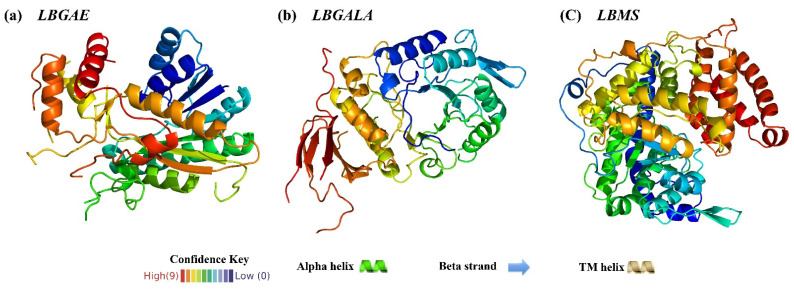
Predicted 3D structure model of protein *LBGAE* (**a**), *LBGALA* (**b**) and *LBMS* (**c**). Green: Alpha helix, lines and arrows: Beta strand. Brown: Helix. Red to blue representative confidence key from high to low.

**Figure 3 plants-10-00321-f003:**
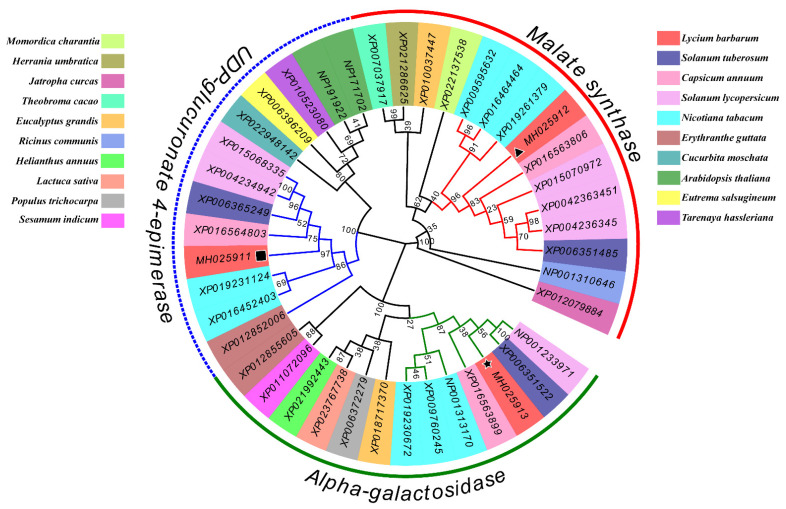
Phylogenetic tree based on the genes: *LBGAE*, *LBGALA* and *LBMS*, with another 20 species by neighbor-joining analysis. Numbers of the branches indicate relative bootstrap support (1000 replications). ★: *LBGALA* ▲: *LBMS* ■: *LBGAE*.

**Figure 4 plants-10-00321-f004:**
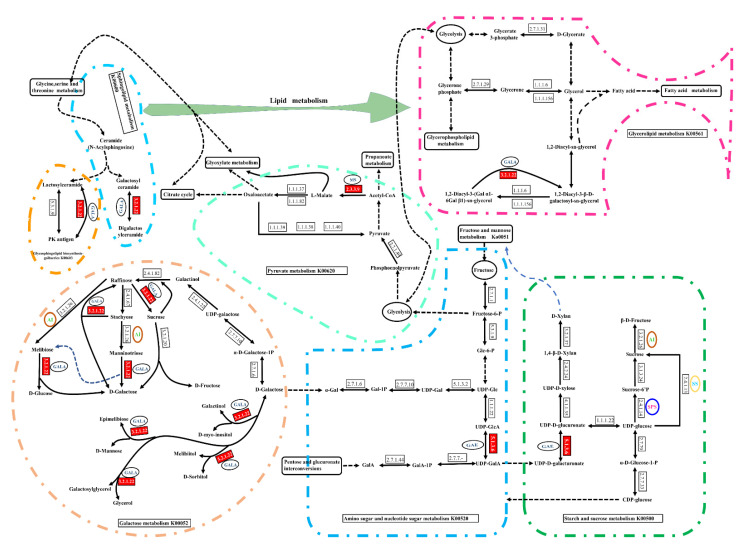
Schematic representation of involvement of genes *LBGAE* (*Lycium barbarum* UDP-glucuronate 4-epimerase), *LBGALA* (*Lycium barbarum* alpha-galactosidase), and *LBMS* (*Lycium barbarum* malate synthase) in the regulating of four sugar metabolism and lipid metabolism pathways. Red representative gene significantly upregulated.

**Figure 5 plants-10-00321-f005:**
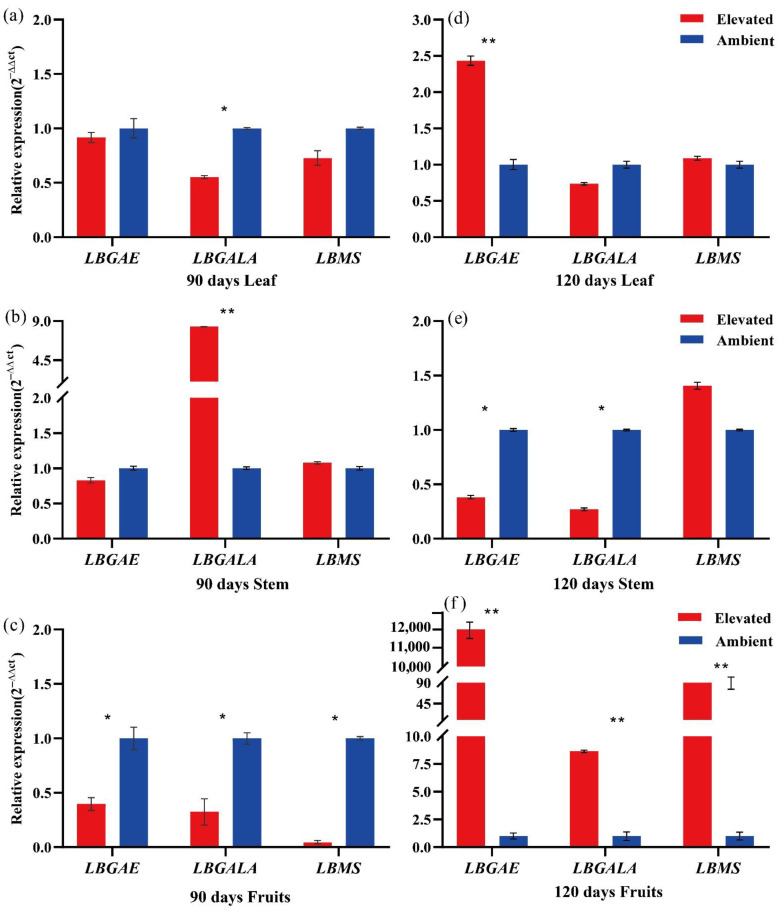
Relative expression of the genes *LBGAE*, *LGALA* and *LBMS* in leaves (**a**,**d**), stems (**b**,**e**), and fruits (**c**,**f**) for 90 days and 120 days under ambient and elevated CO_2_ conditions. Red and blue bars represent elevated and ambient CO_2_ concentration treatment, respectively. * *p* < 0.05, ** *p* < 0.01.

**Figure 6 plants-10-00321-f006:**
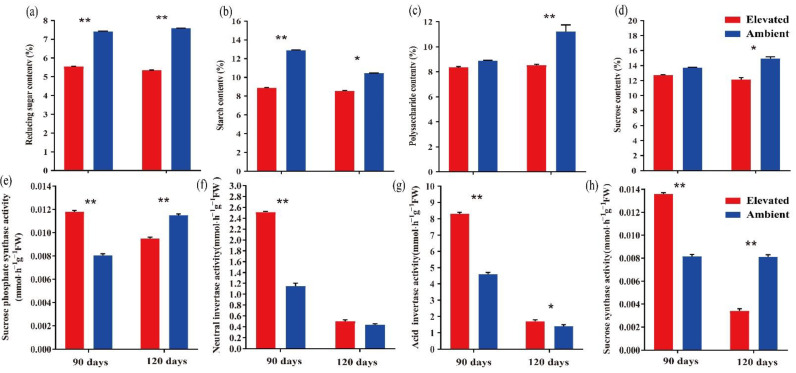
The content of reducing sugar (**a**), starch (**b**), polysaccharide (**c**), sucrose (**d**) and activity of sucrose metabolism-related enzymes sucrose phosphate synthase (**e**), neutral invertase (**f**), acid invertase (**g**), and sucrose synthase (**h**) of goji berry fruit samples grown under ambient and elevated CO_2_ for 90 and 120 days. Red and blue bars represent elevated and ambient CO_2_ concentration treatment, respectively. ** *p* < 0.01, * *p* < 0.05.

**Table 1 plants-10-00321-t001:** Analysis of protein characteristics of genes *LBGAE*, *LBGALA*, *LBMS*.

Gene	Sequence Length (bp)	Protein Length (aa)	Domains and Position	Physicochemical Properties					Secondary Structure			Subcellular Location
				Molecular Weight (kDa)	Theoretical pI	Instability Index	Aliphatic Index	Grand Average of Hydrophilicity	Alpha Helix (%)	Random Coil (%)	Extended Strand (%)	
*LBGAE* (MH025911)	1837	411	SDR family (73–401)	55.65	10.19	22.96	40.27	−1.019	45.99	39.17	14.84	mitochondria
*LBGALA* (MH025913)	1233	571	AmyAc family (38–409)	45.04	5.32	32.62	71.90	−0.369	29.27	53.66	17.07	chloroplast
*LBMS* (MH025912)	2449	410	MS (19–571)	65.00	8.45	40.23	81.65	−0.358	48.51	41.16	10.33	peroxisome

Note: SDR: short-chain dehydrogenases/reductases; AmyAc: Alpha-amylase catalytic; MS: malate synthase.

## Data Availability

Data are contained within the article or Appendix A.

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
