# Peer review of "Cloning and Characterization of Three Sugar Metabolism Genes (*LBGAE*, *LBGALA*, and *LBMS*) Regulated in Response to Elevated CO_2_ in Goji Berry (*Lycium barbarum* L.)"

_plants, 2021, doi:10.3390/plants10020321_

Round 1

Reviewer 1 Report

COMMENTS TO THE AUTHORS:
In this manuscript, the authors cloned full-length cDNA of three sugar metabolism genes LBGAE (UDP-glucuronate 4-epimerase), LBGALA (alpha-galactosidase), and LBMS (malate synthase) that were previously identified responding to eCO2. Sugar metabolism enzymes regulated by these genes were also investigated and their activity varied significantly from ambient to eCO2 in 90 days and 120 days samples along with sugars. This research was reasonably designed and well presented. I have a few suggestions for its further improvement.

1: The major issue is the resolution of Figure 1, especially the two enlarged insert on the top left of Figure 1a and bottom left of Figure 1. The contents are unreadable at all.

2: In the Abstract, line 17, the word “However” seems not proper for using here. Please re-phrase.

3: In the Abstract, lines 28-29 “The study also discussed ……. sugar metabolism pathway”, there is no meaningful information provided. Either deleted it or re-phrase it.

4: The methods for generating Figure 2 is not clear. Please add more related information. Furthermore, the legend of Figure 2 is not sufficient to elaborate this complicated illustration. Please explain more. For example, the different color dotted line shapes, different colored boxes, etc.

5:  In page 12, line 410, replace “Ha, Ma, Cao, Guo and Song” with “Ha et al”. Similarly, change page 13, line 412.

6: In page 13, lines 426-428, for functional characterization of these three genes, knock-out or knock down mutants will be way more important than overexpression study. Please add such information in the paper.

7: Double check the References to make them consistent. For example, some journal article titles (as well as journal names) are Capitalized and some are not; in addition, the journal name for ref16 is abbreviated.

Author Response

Reviewer 1

Comments and corrections are included in the attached manuscript.

The authors agree with the general comments of the Reviewer. Thank you for the suggestions. Please see the response is listed as follows.

1

The major issue is the resolution of Figure 1, especially the two enlarged insert on the top left of Figure 1a and bottom left of Figure 1. The contents are unreadable at all.

Figure 1 a and b were replaced by a high-pixel graphs, and can be read very clearly with enlarged dimensions

2

In the Abstract, line 17, the word “However” seems not proper for using here. Please re-phrase.

The word “However” was deleted and rephrased the sentence. In order to understand the regulatory mechanisms and improve the quality of fruit in the changing climate, it is essential to characterize sugar metabolism genes that respond to eCO2.

3

In the Abstract, lines 28-29 “The study also discussed ……. sugar metabolism pathway”, there is no meaningful information provided. Either deleted it or re-phrase it.

In lines 28-29, The sentences “The study also discussed the involvement of the three genes in different sugar metabolism path-ways.” was deleted.

4

The methods for generating Figure 2 is not clear. Please add more related information. Furthermore, the legend of Figure 2 is not sufficient to elaborate this complicated illustration. Please explain more. For example, the different color dotted line shapes, different colored boxes, etc.

The method for Figure 2 was added in the line 402-404, “Protein tertiary structure was modeled by the homology-modeling server SWISS-MODEL (http://swissmodel.expasy.org/), and the models were further analyzed and refined using Swiss-Pdb Viewer v4.1.0.”.

Also, the information was added to Figure 2 to indicate the composition of the tertiary structure and the significance of the color, and illustrated in the legend. “Predicted 3D structure model of protein LBGAE (a), LBGALA (b) and LBMS (c). Green: Alpha helix, lines and arrows: Beta strand, Brown: Helix. Red to blue representative confidency key from high to low.”

5

In page 12, line 410, replace “Ha, Ma, Cao, Guo and Song” with “Ha et al”. Similarly, change page 13, line 412.

We replaced it with Ha et al., and Liu et al., in lines 437 and 439.

6

In page 13, lines 426-428, for functional characterization of these three genes, knock-out

or knock down mutants will be way more important than overexpression study. Please add such information in the paper.

The sentence was changed in 454 line: “a detailed study on the role of three genes and their mechanism in regulating sugar metabolism by knock-out or knock down mutants is needed.”

7

Double check the References to make them consistent. For example, some journal article titles (as well as journal names) are Capitalized and some are not; in addition, the journal name for ref16 is abbreviated.

We revised all the references according to Plants journal instructions.

Reviewer 2 Report

Line 213, 214, 234  correct *p<0.05,  **p<0.01

Line 222 correct (p<0.05),

Line 223 correct ambient CO2 conditions.

Line 227, 229 correct (p<0.01).

Line 256, 257 correct (Figure 4)

Line 317 correct observed (Figure 6 e, f, g, h).

Line 341 correct period of 140-160 days.

Line 353 correct stored at -20 ℃

Line 367, 367 correct of 94 ℃ for the 40 s, 58 ℃ for 30 s, 72 ℃ for 1 min 20 s, and then a final extension at 72 ℃ for 10 min and stored at 16 ℃.

Line 398 correct 0.4 μl of each Primer

Line 401 correct of 94 ℃ for 15 s, 60 ℃ for 1 min, and stored at 16 ℃.

Line 466 correct different CO2

Line 471 correct in Glycine max.

Line 494 correct Effects of doubled atmospheric CO2 concentration on sugar accumulation of different

Line 502-504 correct Expression patterns, activities and sugar metabolism regulation of sucrose phosphate synthase, sucrose synthase, neutral invertase and soluble acid invertase in different Goji cultivars during fruit development.

Line 508-509 correct Effects of hight atmospheric CO2 concentrations on activities of sucrose metabolism-related enzymes in Lycium barbarum fruit.

Line 510 correct in Arabidopsis thaliana

Author Response

Reviewer 2

Comments and corrections are included in the attached manuscript.

The authors appreciate reviewers time and suggestions. We reflected all changes in the manuscript. Please see the response below,

1

Line 213, 214, 234  correct *p<0.05,  **p<0.01

We corrected P value

2

Line 222 correct (p<0.05),

The P value was corrected as suggested

3

Line 223 correct ambient CO2 conditions.

We made suggested changed in line 239

4

Line 227, 229 correct (p<0.01).

We corrected P value.

5

Line 256, 257 correct (Figure 4)

We made changes in line 273

Line 317 correct observed (Figure 6 e, f, g, h).

We made corrections in line 336

Line 341 correct period of 140-160 days.

We modified the period in line 260

Line 353 correct stored at -20 ℃

We changed the storage temperature in line 372

Line 367, 367 correct of 94 ℃ for the 40 s, 58 ℃ for 30 s, 72 ℃ for 1 min 20 s, and then a final extension at 72 ℃ for 10 min and stored at 16 ℃.

We made all suggested changes (lines 389-390)

Line 398 correct 0.4 μl of each Primer

The corrections were made in line 422

Line 401 correct of 94 ℃ for 15 s, 60 ℃ for 1 min, and stored at 16 ℃.

Thermal cycle temperature and timings were corrected in line 424-425 as suggested.

Line 466 correct different CO2

Line 471 correct in Glycine max.

Line 494 correct Effects of doubled atmospheric CO2 concentration on sugar accumulation of different

Line 502-504 correct Expression patterns, activities and sugar metabolism regulation of sucrose phosphate synthase, sucrose synthase, neutral invertase and soluble acid invertase in different Goji cultivars during fruit development.

Line 508-509 correct Effects of hight atmospheric CO2 concentrations on activities of sucrose metabolism-related enzymes in Lycium barbarum fruit.

Line 510 correct in Arabidopsis thaliana

We revised all the references according to Plants journal instructions.

We made all changes in the manuscript as suggested by the reviewer.